# The Development of a Multiplex Real-Time Quantitative PCR Assay for the Differential Detection of the Wild-Type Strain and the MGF505-2R, EP402R and I177L Gene-Deleted Strain of the African Swine Fever Virus

**DOI:** 10.3390/ani12141754

**Published:** 2022-07-08

**Authors:** Kang Zhao, Kaichuang Shi, Qingan Zhou, Chenyong Xiong, Shenglan Mo, Hongjin Zhou, Feng Long, Haina Wei, Liping Hu, Meilan Mo

**Affiliations:** 1College of Animal Science and Technology, Guangxi University, Nanning 530005, China; zhaokang0519@163.com (K.Z.); xiongcy0509@163.com (C.X.); zhouw1714@163.com (H.Z.); 2Guangxi Center for Animal Disease Control and Prevention, Nanning 530001, China; zhouqingan1@163.com (Q.Z.); moshl_2015@126.com (S.M.); longfeng1136@163.com (F.L.); weihaina@sina.cn (H.W.); hu.liping@foxmail.com (L.H.)

**Keywords:** African swine fever virus (ASFV), multiplex real-time qPCR, differentiation, detection, wild-type strain, gene-deleted strain

## Abstract

**Simple Summary:**

African swine fever (ASF) was first reported in August 2018 in China, and the naturally gene-deleted ASFV strain was first identified in 2020 in this country. The vaccine candidates that deleted some virulent genes from the virulent parental strains have also been reported in many countries. To differentiate the wild-type and gene-deleted ASFV strains, four pairs of specific primers and TaqMan probes targeting the ASFV B646L (p72), I177L, MGF505-2R and EP402R (CD2v) genes were designed. After optimizing the reaction conditions, a multiplex real-time qPCR assay for the differential detection of the wild-type and gene-deleted ASFV strains was developed. The assay was further used to test 4239 clinical samples, and 534 samples were positive for ASFV, of which 30 samples lacked B646L, I177L, MGF505-2R and/or EP402R genes. The assay showed high specificity, sensitivity and repeatability, and it provided a reliable method for evaluating ASFV in clinical samples.

**Abstract:**

African swine fever virus (ASFV) causes African swine fever (ASF), a devastating hemorrhagic disease of domestic pigs and wild boars. Currently, the MGF505R, EP402R (CD2v) and I177L gene-deleted ASFV strains were confirmed to be the ideal vaccine candidate strains. To develop an assay for differentiating the wild-type and gene-deleted ASFV strains, four pairs of specific primers and TaqMan probes targeting the ASFV B646L (p72), I177L, MGF505-2R and EP402R (CD2v) genes were designed. A multiplex real-time qPCR assay for the differential detection of the wild-type and gene-deleted ASFV strains was developed after optimizing the reaction conditions, including the annealing temperature, primer concentration and probe concentration. The results showed that the multiplex real-time qPCR assay can specifically test the ASFV B646L (p72), I177L, MGF505-2R and EP402R (CD2v) genes with a limit of detection (LOD) of 32.1 copies/μL for the B646L (p72) gene, and 3.21 copies/μL for the I177L, MGF505-2R and EP402R (CD2v) genes. However, the assay cannot test for the classical swine fever virus (CSFV), porcine reproductive and respiratory syndrome virus (PRRSV), porcine epidemic diarrhea virus (PEDV), porcine deltacoronavirus (PDCoV), porcine circovirus type 2 (PCV2), PCV3 and pseudorabies virus (PRV). The assay demonstrated good repeatability and reproducibility with coefficients of variation (CV) less than 1.56% for both the intra- and inter-assay. The assay was used to test 4239 clinical samples, and the results showed that 12.60% (534/4239) samples were positive for ASFV, of which 10 samples lacked the EP402R gene, 6 samples lacked the MGF505-2R gene and 14 samples lacked the EP402R and MGF505-2R genes. The results indicated that the multiplex real-time qPCR developed in this study can provide a rapid, sensitive and specific diagnostic tool for the differential detection of the ASFV B646L, I177L, MGF505-2R and EP402R genes.

## 1. Introduction

African swine fever (ASF), which is caused by the ASF virus (ASFV), is an acute, hemorrhagic and infectious disease of domestic pigs and wild boars [1]. The highly virulent ASFV can cause acute ASF with nearly 100% mortality [2]. ASF was first identified in Kenya in 1921, in Europe in 1957 and in the Caucasus region and southern Russia in 2007 [3,4]. ASF has been reported in many Asian countries, including China, Korea, Mongolia, Vietnam, Cambodia, Laos, the Philippines and Indonesia, since 2018 [5,6]. ASF was recently reported again in the Dominican Republic and Haiti in 2021 almost 40 years after the last outbreak in these countries [7,8]. Currently, ASF is still an epidemic in some countries in Africa, Europe and Asia and pose a huge challenge to pork production [9,10]. ASF has caused huge economic losses to the swine industry worldwide since the 1920s and is listed as a notifiable disease to the World Organisation for Animal Health (OIE).

The ASFV belongs to the family *Asfarviridae*, genus *Asfivirus* [11]. ASFV is an enveloped, double-stranded DNA virus with 170 to 193 kb genome, which encodes more than 168 kinds of proteins [12]. In recent times, twenty-four genotypes of ASFV strains were identified in different countries according to the sequences of the partial B646L (p72) gene [13]. To date, only genotypes I and II of ASFV strains have been reported outside Africa [14]. The ASFV was introduced into China in August 2018 [5] and led to a sharp reduction in pigs raised in China within a year [15]. In China, genotype II ASFV strains are the main circulating strains in the field, and genotype I ASFV strains have also been reported [16,17].

Vaccination with effective vaccines is one of the most useful strategies for preventing and controlling ASF. However, no effective commercial vaccine could be used for ASF in the field until May 2022. Recently, several highly effective and safe vaccine candidate strains with gene-deletion were reported [18,19]. For example, a recombinant ASFV vaccine candidate strain with the deletion of 121 nucleotides in the I177L gene has shown to protect pigs against the highly pathogenic ASFV isolates that are presently circulating in Europe and Asia [20,21,22]. In June 2022, the attenuated vaccine using the ASFV strain with 121 bp deletion in the I177L gene was officially approved for marketing in Vietnam to prevent ASF, which was the first commercial vaccine approved for clinical application in the world (http://link.gov.vn/vtUM759t, accessed on 8 June 2022). In addition, the MGF360-505R and EP402R gene-deleted vaccine, which has been developed in recent years, has become one of the ideal vaccine candidates [23,24,25,26]. These vaccine candidates are the most promising for commercial applications in the field in the future. Furthermore, natural MGF360-505R and EP402R (CD2v) gene-deleted ASFV strains, which show decreased virulence with lower morbidity and mortality, have been identified in several provinces in China [27]. Consequently, it is necessary to develop a rapid, specific and sensitive assay to differentiate the wild-type and gene-deleted ASFV strains.

The real-time quantitative polymerase chain reaction (real-time qPCR) offers great advantages for the detection and quantification of the targeted genome with high specificity, sensitivity and repeatability [28]. Currently, several real-time qPCR assays for the detection of ASFV have been reported. Single real-time qPCR assays have been developed for the detection of the wild-type ASFV strains [29,30,31,32] and of the MGF-360-12L, UK or I177L gene-deleted ASFV strains [33]. Multiplex real-time qPCR assays have been developed for the detection of the wild-type ASFV strains [34,35] and of the gene-deleted ASFV strains which lacked the MGF505-2R gene [36], the MGF505-2R and EP402R genes [37]. However, no assay for the simultaneous detection of the ASFV strains which lacks the MGF505-2R, EP402R and I177L genes has been reported until now. In this study, we developed a multiplex real-time qPCR assay for the differential detection of the wild-type ASFV strains and of the MGF505-2R, EP402R and I177L gene-deleted ASFV strains.

## 2. Materials and Methods

### 2.1. Viral Strains

The vaccine strains of CSFV (C vaccine strain), PRRSV (TJM-F92 vaccine strain), porcine circovirus type 2 (PCV2, SX07 vaccine strain), porcine epidemic diarrhea virus (PEDV, SCJY-1 vaccine strain) and pseudorabies virus (PRV, Bartha-K61 vaccine strain) were purchased from Huapai Bioengineering Group Co., Ltd. (Chengdu, China). The clinical positive samples of ASFV, PCV3 and porcine deltacoronavirus (PDCoV) were provided by the Guangxi Center for Animal Disease Control and Prevention (CADC), China. These viruses were stored at −70 °C until they were used.

### 2.2. Clinical Samples

From January 2021 to December 2021, a total of 4239 clinical samples were collected from different pig farms, slaughterhouses, farmers’ markets and harmless disposal sites in Guangxi province, southern China. The samples included environmental swabs, nasopharyngeal swabs, whole blood and tissue samples (kidneys, spleens and lymph nodes). The samples were transported to our laboratory at conditions of 0–4 °C immediately after collection and were stored at −70 °C until they were used.

### 2.3. DNA Extraction

The homogenized material from the tissue samples, including kidneys, spleens and lymph nodes, was obtained using a blender (TIANLONG Scientific, Xi’an, China), freeze-thawed 3 times and centrifuged at 10,000× *g* for 10 min at 4 °C. The supernatants were used for total nucleic acids extraction.

Each environmental sample (swab) and nasopharyngeal swab was diluted in a tube with 1.0 mL PBS (pH 7.2), vortexed for 10 s and centrifuged at 10,000× *g* for 5 min at 4 °C. The supernatants were used for total nucleic acids extraction.

The anticoagulant whole blood samples with EDTA-Na_2_ (15 g/L) were vortexed for 10 s and centrifuged at 10,000× *g* for 5 min at 4 °C. The supernatants were used for total nucleic acids extraction.

The total nucleic acids were extracted from 200 μL of the supernatants from the clinical samples using the GeneRotex 96 Automatic Nucleic Acid Extractor (TIANLONG Scientific, Xi’an, China) with Viral DNA/RNA Isolation Kit 4.0 (TIANLONG Scientific, Xi’an, China) according to the manufacturer’s instructions. The extracted nucleic acids were tested for ASFV by the developed multiplex real-time qPCR using the QuantStudio 6 qPCR system (ABI, Carlsbad, CA, USA).

### 2.4. Primers and Probes

Four pairs of specific primers and corresponding TaqMan probes were designed using the Primer Express Software v3.0 (ABI, Los Angeles, CA, USA), which targeted the conserved regions of the ASFV B646L gene, MGF505-2R gene, EP402R gene and I177L gene. The sequences of the primers and probes are shown in Table 1.

### 2.5. Construction of Standard Plasmids

Using ASFV DNA as a template, the targeted fragments of the B646L gene, MGF505-2R gene, EP402R gene and I177L gene were amplified by PCR, purified, ligated to the pMD18-T vector (TaKaRa, Beijing, China) and then transformed into *E. coli* DH5α competent cells. The positive clones were selected and grown at 37 °C for about 20 h. The recombinant plasmids were extracted, named as pASFV-B646L, pASFV-MGF505-2R, pASFV-EP402R and pASFV-I177L, respectively, and used as standard plasmids for developing the multiplex real-time qPCR. A NanoDrop spectrophotometer (Thermo Fisher, Waltham, MA, USA) was used to quantify the ultraviolet absorbance of the standard plasmids at 260 nm and 280 nm. The following formula was used to determine their concentrations: plasmid copy number (copies/μL) = (plasmid concentration × 10^−9^ × 6.02 × 10^23^)/(660 Dalton/bases × DNA length).

The sequences at both ends of the ASFV MGF505-2R, EP402R and I177L genes were artificially synthesized by TAKARA Biomedical Technology Co. Ltd. (TaKaRa, Beijing, China) and were cloned into the pMD18-T vector (TaKaRa, Beijing, China) to formulate synthetic plasmids without the MGF505-2R gene (pASFV-ΔMGF505-2R), EP402R gene (pASFV-ΔEP402R) and I177L gene (pASFV-ΔI177L). They were used as the MGF505-2R, EP402R and I177L gene-deleted controls. The sequences of pASFV-ΔMGF505-2R, pASFV-ΔEP402R and pASFV-ΔI177L are shown in Appendix A.

### 2.6. Optimization of the Reaction Conditions of the Multiplex Real-Time qPCR

The multiplex real-time qPCR was performed using the QuantStudio 6 qPCR system (ABI, Carlsbad, CA, USA). The annealing temperature and the concentrations of the primers and probes were determined by the following amplification parameters: 95 °C for 2 min, 40 cycles of 95 °C for 5 s and 56 °C for 30 s. To determine the optimal reaction conditions for the multiplex real-time qPCR, the 20 μL of the fundamental systems was made as follows: 10 μL Premix Ex Taq (Probe qPCR) (TaKaRa, Beijing, China), 2 μL mixtures of the four standard plasmids (each with final concentration 10^8^ copies/μL), 4 pairs of the primers and probes (each with different final concentration) and distilled water. The recombinant plasmids pASFV-B646L, pASFV-MGF505-2R, pASFV-EP402R and pASFV-I177L were used as templates. The different annealing temperatures (52 °C, 54 °C, 56 °C, 58 °C, 60 °C and 62 °C), primer concentrations (0.1 pmol/μL, 0.2 pmol/μL, 0.3 pmol/μL, 0.4 pmol/μL and 0.5 pmol/μL) and probe concentrations (0.1 pmol/μL, 0.2 pmol/μL, 0.3 pmol/μL, 0.4 pmol/μL and 0.5 pmol/μL) were used for amplification by an arrangement and combination test to obtain the optimal reaction conditions. At the end of each cycle, the fluorescent signals were determined.

### 2.7. Construction of Standard Curves

The standard plasmids pASFV-B646L, pASFV-MGF505-2R, pASFV-EP402R and pASFV-I177L were mixed together, serially diluted 10-fold from 3.21 × 10^8^ to 3.21 × 10^2^ copies/μL (final concentration) and used as templates to generate the standard curves of the developed multiplex real-time qPCR.

### 2.8. Specificity Analysis of the Multiplex Real-Time qPCR

The DNA or cDNA of the ASFV, PRRSV, CSFV, PCV2, PCV3, PEDV, PDCoV and PRV, as well as the recombinant gene-deleted plasmids pASFV-ΔMGF505-2R, pASFV-ΔEP402R and pASFV-ΔI177L, were used to evaluate the specificity of the developed assay. The sterilized distilled water was used as the negative control.

### 2.9. Sensitivity Analysis of the Multiplex Real-Time qPCR

The standard plasmids pASFV-B646L, pASFV-MGF505-2R, pASFV-EP402R and pASFV-I177L were mixed together, serially diluted 10-fold from 3.21 × 10^8^ to 3.21 × 10^−1^ copies/μL (final concentration) for each plasmid and used as templates to evaluate the sensitivity of the developed assay.

### 2.10. Repeatability Analysis of the Multiplex Real-Time qPCR

The standard plasmids pASFV-B646L, pASFV-MGF505-2R, pASFV-EP402R and pASFV-I177L were mixed and 10-fold serially diluted. Three final concentrations of 3.21 × 10^8^, 3.21 × 10^6^ and 3.21 × 10^4^ copies/μL for each plasmid were used to run the intra- and inter-assay tests. The coefficients of variation (CVs) were used to evaluate the repeatability and the reproducibility of the assay.

### 2.11. Test of Clinical Samples by the Multiplex Real-Time qPCR

The developed multiplex real-time qPCR assay was used to test 4239 clinical samples collected in Guangxi province, southern China, from January 2021 to December 2021.

At the same time, the above samples were tested for ASFV by the OIE-recommended real-time qPCR for ASFV [38] with a total volume of 25 μL, as shown in Table 2. The amplification was performed at 50 °C for 2 min and 95 °C for 10 min and then with 40 cycles of 95 °C for 15 s and 58 °C for 60 s, and the fluorescent signals at the end of each cycle were determined.

The coincidence rate between the developed multiplex real-time qPCR and the OIE-recommended real-time qPCR was evaluated.

### 2.12. Statistical Analysis

The statistical analyses were conducted by regression analysis, the Pearson’s correlation coefficient and an analysis of variance (ANOVA) using the GraphPad Prism version 7.04 software (GraphPad Software, Santiago, CA, USA) for linear regression and for the tested results of clinical samples.

## 3. Results

### 3.1. Construction of Standard Plasmids

The targeted fragments of the ASFV B646L gene, MGF505-2R gene, EP402R gene and I177L gene were amplified by PCR, purified and ligated to the pMD18-T vector (TaKaRa, Beijing, China), and they were then transferred to *E. coli* DH5α competent cells (TaKaRa, Beijing, China). Positive clones were cultivated, and the plasmid constructs were extracted and quantified. The results showed that the initial concentrations of the standard plasmids named pASFV-B646L, pASFV-MGF505-2R, pASFV-EP402R and pASFV-I177L were 4.51 × 10^10^ copies/μL, 3.21 × 10^10^ copies/μL, 3.86 × 10^10^ copies/μL and 5.16 × 10^10^ copies/μL, respectively.

### 3.2. Optimal Reaction Conditions of the Multiplex Real-Time qPCR

The standard plasmids were used to optimize the multiplex real-time qPCR reaction parameters, including the annealing temperature and the primer and probe concentrations using an arrangement and combination test.

The developed multiplex real-time qPCR (20 μL) contained 10 μL of Premix Ex Taq (TaKaRa, Beijing, China), P72-F/R (forward and reward primer) (20 pmol/μL) 0.2 μL each, P72-P probe (20 pmol/μL) 0.2 μL, I177L-F/R (forward and reward primer) (20 pmol/μL) 0.1 μL each, I177L-P probe (20 pmol/μL) 0.1 μL, EP402R-F/R (forward and reward primer) (20 pmol/μL) 0.2 μL each, EP402R-P probe (20 pmol/μL) 0.2 μL, MGF505-2R-F/R (forward and reward primer) (20 pmol/μL) 0.2 μL each, MGF505-2R-P probe (20 pmol/μL) 0.3 μL, 2 μL of total DNA/RNA and 5.8 μL of sterilized distilled water (Table 2). The amplification was performed at 95 °C for 10 s and then with 40 cycles of 95 °C for 5 s, 56 °C for 30 s, and the fluorescent signals at the end of each cycle were determined. A sample with a Ct value of ≤35 cycles was judged as positive, whereas a sample with a Ct value of >35 cycles was judged as negative.

### 3.3. Standard Curves of the Multiplex Real-Time qPCR

The multiplex real-time qPCR standard curves were generated using 10-fold serially diluted standard plasmids from 3.21 × 10^8^ to 3.21 × 10^2^ copies/μL (final concentration). The results showed that the corresponding slope of the equation, correlation coefficient (R^2^) and amplification efficiency (E) were as follows: −3.211, 0.999 and 104.9% for the B646L gene; −3.159, 0.999 and 107.3% for the I177L gene; −3.152, 0.999 and 107.6% for the EP402R gene; and −3.115, 0.998 and 109.4% for the MGF505-2R gene (Figure 1).

### 3.4. Specificity of the Multiplex Real-Time qPCR

The multiplex real-time qPCR assay was validated for specificity using DNA/cDNA from the ASFV, PRRSV, CSFV, PCV2, PCV3, PEDV, PDCoV and PRV and using recombinant plasmids pASFV-ΔMGF505-2R, pASFV-ΔEP402R and pASFV-ΔI177L. The results showed that only the ASFV generated positive amplification curves for the four fluorescence channels of FAM, VIC, Cy5 and Texas Red corresponding to the B646L gene, MGF505-2R gene, EP402R gene and I177L gene, respectively, whereas other viruses and plasmids did not generate amplification curves, indicating that the developed multiplex real-time qPCR could specifically test the B646L gene, MGF505-2R gene, EP402R gene and I177L gene of the ASFV (Figure 2).

### 3.5. Sensitivity of the Multiplex Real-Time qPCR

The standard plasmids pASFV-B646L, pASFV-MGF505-2R, pASFV-EP402R and pASFV-I177L were serially diluted 10-fold from 3.21 × 10^8^ to 3.21 × 10^−1^ copies/μL (final concentration) and were used to determine the limit of detection (LOD) of the multiplex real-time qPCR. The results showed that the LOD of pASFV-B646L was 32.1 copies/μL, and the LOD of pASFV-MGF505-2R, pASFV-EP402R and pASFV-I177L was 3.21 copies/μL (Figure 3).

### 3.6. Repeatability of the Multiplex Real-Time qPCR

The repeatability and reproducibility of the developed multiplex real-time qPCR was evaluated using three concentrations of the standard plasmids: 3.21 × 10^8^, 3.21 × 10^6^ and 3.21 × 10^4^ copies/μL (final concentration). The results showed that the coefficients of variation (CVs) of the intra- and inter-assay were from 0.26% to 0.92% and from 0.53% to 1.56%, respectively (Table 3).

### 3.7. Evaluation of Clinical Samples

A total of 4239 clinical samples were collected from January 2021 to December 2021 in Guangxi province, southern China, and were tested by the developed multiplex real-time qPCR. The results showed that 12.60% (534/4239) of samples were positive for the ASFV (Table 4). Out of the different sample types, the harmless disposal sites showed the highest positive rate of 28.34% (121/427), followed by the slaughterhouses with 17.30% (330/1980), the farmers’ markets with 10.70% (67/626) and the breeding farms with the lowest positive rate of 1.25% (15/1278). Additionally, the results showed that, of the 534 positive samples, 10 samples lacked the EP402R gene, 6 samples lacked the MGF505-2R gene, 14 samples lacked the EP402R and MGF505-2R genes and no sample lacked the I177L gene. Three different kinds of gene-deleted clinical samples are shown in Figure 4.

The clinical samples were also tested by the OIE-recommended real-time qPCR targeting the ASFV B646L gene [38]. The results of the developed multiplex real-time qPCR were consistent with the results of the OIE-recommended real-time qPCR. The coincidence rate of these two assays was 100%, suggesting good consistency between these two methods while using the B646L gene as a target.

## 4. Discussion

Currently, ASF is still an epidemic in Africa, the trans-Caucasus region, eastern Europe, the Russian Federation and Asia, which poses a great challenge to the pig industry and port trade all over the world [9,10]. At present, there is no completely effective way to prevent and control ASF, and strict biosafety and culling of the infected pigs are common measures to reduce the losses of this disease. The development of safe and effective vaccines is one of the key strategies. In order to generate attenuated ASF vaccines, virulent ASFV strains have had several virulence-associated genes deleted, such as I177L, 9GL, UK, EP402R (CD2v), DP148R, A238L, MGF505, MGF360, etc., and have shown decreased virulence and high efficacy to protect pigs infected with their parental strains [24,39,40]. Scientists from China and the United Kingdom have reported that the ASFV strains, which were deleted the MGF505 and MGF360 genes or the MGF505, MGF360 and EP402R (CD2v) genes, greatly attenuated their virulence and protected pigs from challenges with their parental trains [24,41]. Scientists from the USA and Vietnam have reported that the ASFV strain with the I177L gene deletion can also provide homology protection [20,21,22]. Furthermore, the naturally gene-deleted ASFV strains of MGF360-505R and EP402R genes, which show decreased virulence with lower morbidity and mortality, were identified in several provinces in China [27]. Therefore, to differentiate the wild-type strains and the gene-deleted strains, including the naturally gene-deleted strains and the man-made gene-deleted vaccine candidate strains, a rapid, sensitive and specific assay is necessary to differentially detect these ASFV strains.

The qRT-PCR/PCR is a rapid, specific, sensitive and accurate method for the detection of viral nucleic acids and can be conveniently used for the detection and quantification of viral swine pathogens [42,43]. In this study, a quadruplex real-time qPCR based on four pairs of specific primers and corresponding probes was developed, and it can be used for the differential detection of the wild-type and gene-deleted ASFV strains in one reaction. The assay showed high sensitivity with an LOD from 3.21 copies/μL to 32.1 copies/μL for different templates and can specifically test the ASFV B646L, I177L, MGF505-2R and EP402R genes without cross-reaction with other porcine viruses. Using this multiplex real-time qPCR assay, the ASFV strain with the B646L gene can be tested regardless of whether it is the wild-type or gene-deleted ASFV. The wild-type ASFV strain shows the simultaneous presence of these four genes, and the gene-deleted ASFV strain shows the deletion of one or more than one gene of the I177L, MGF505-2R and EP402R genes. The assay was verified for its practicality and usefulness by testing 4239 clinical samples in this study.

Since ASF was first reported in China in August 2018, this disease has spread rapidly throughout all provinces in mainland China in a short time [6,44]. In this study, a total of 4239 clinical samples form Guangxi province, southern China, were tested by the developed multiplex real-time qPCR. The results showed that 12.60% of samples were positive for ASFV, of which the positive rates of the harmless disposal sites, slaughterhouses, farmers’ markets and breeding farms were 28.34%, 17.30%, 10.70% and 1.25%, respectively, indicating that the ASFV was still a serious epidemic in Guangxi province, southern China. Of the 534 positive samples, 285 (53.37%) were environmental samples, which came from the harmless disposal sites, slaughterhouses, farmers’ markets and breeding farms. Since ASFV can survive for a long period in the environment [45], the viruses easily spread from these places through moving humans and vehicles. Therefore, it is very important to clean and disinfect the environments of these places, especially harmless disposal sites, to cut off the transmission of ASFV.

It is noteworthy that the gene-deleted ASFV strains are circulating in the field. Among the 534 ASFV-positive samples, 10 samples lacked the EP402R gene, 6 samples lacked the MGF505-2R gene, 14 samples lacked the EP402R and MGF505-2R genes and no sample lacked the I177L gene. This indicates that the wild-type and gene-deleted ASFV strains were simultaneously circulating in the field in Guangxi province, which further verified the existence of different gene-deleted ASFV strains in China [27]. They may be natural variations, or they may come from man-made illegal vaccines, which need to be further investigated and verified. At present, no I177L gene-deleted ASFV strain, which was verified to be an ideal vaccine candidate and has been approved officially to be used in the field as an attenuated vaccine since June 2022 in Vietnam [20], was found in the field in China. However, since the border between China and Vietnam is more than 1300 km, and since personnel and goods exchanges are frequent, the illegal trade of live pigs and pork products between these two countries is inevitable. Therefore, the I177L gene-deleted ASFV strain needs to be continuously monitored at present in the case of the transmission of this strain from Vietnam to China.

## 5. Conclusions

A multiplex real-time qPCR assay was successfully developed for the differential detection of the wild-type and gene-deleted ASFV strains in this study. The assay showed high specificity, sensitivity and repeatability, and it provided a reliable method for evaluating ASFV in clinical samples. The tested results of the clinical samples showed that harmless disposal sites, slaughterhouses and farmers’ markets in Guangxi province, southern China, are severely polluted by ASFV at present, which are important risk points for the prevention and control of ASF.

## Figures and Tables

**Figure 1 animals-12-01754-f001:**
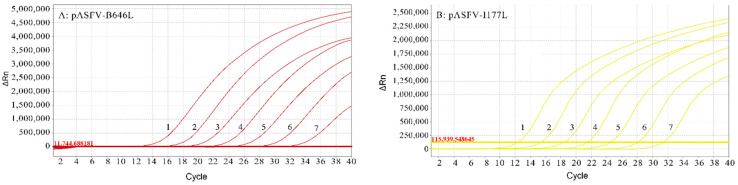
Dynamic curves and standard curves of the multiplex real-time qPCR: The dynamic curves were generated by using the recombinant standard plasmids pASFV-B646L (**A**), pASFVI177L (**B**), pASFV-MGF505-2R (**C**) and pASFVEP402R (**D**). The standard curves (**E**) were generated from the dynamic curves. In (**A**–**D**), the plasmid concentrations of curves 1 to 7 ranged from 3.21 × 10^8^ copies/µL to 3.21 × 10^2^ copies/µL (final concentration).

**Figure 2 animals-12-01754-f002:**
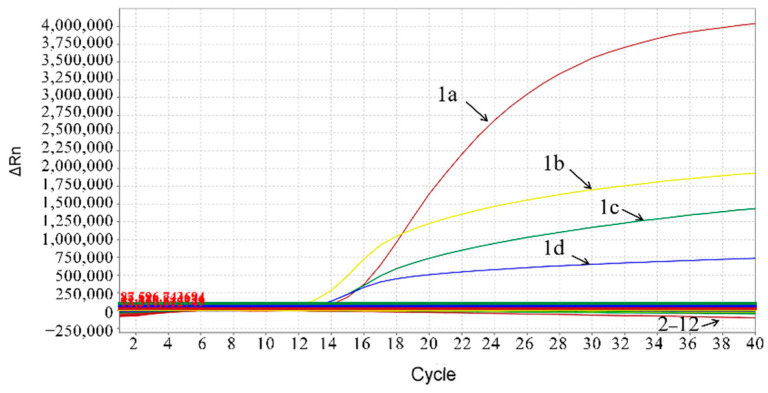
Specificity analysis of the multiplex real-time qPCR: 1a: pASFV-B646L; 1b: pASFV-I177L; 1c: pASFV-MGF505-2R; 1d: pASFV-EP402R; 2: pASFV-ΔI177L; 3: pASFV-ΔMGF505-2R; 4: pASFV-ΔEP402R; 5: CSFV; 6: PRRSV; 7: PCV2; 8: PCV3; 9: PEDV; 10: PDCoV; 11: PRV; 12: Negative control.

**Figure 3 animals-12-01754-f003:**
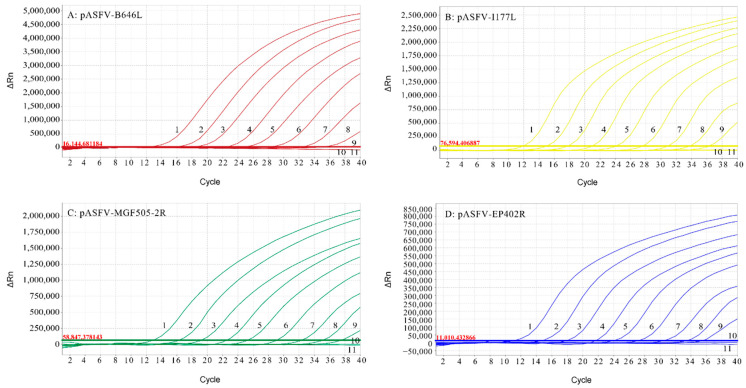
Sensitivity analysis of the multiplex real-time qPCR: The dynamic curves were generated by using the recombinant standard plasmids pASFV-B646L (**A**), pASFVI177L (**B**), pASFV-MGF505-2R (**C**) and pASFVEP402R (**D**). In (**A**–**D**), the plasmid concentrations of curves 1 to 10 ranged from 3.21 × 10^8^ copies/µL to 3.21 × 10^−1^ copies/μL (final concentration); 11: Negative control.

**Figure 4 animals-12-01754-f004:**
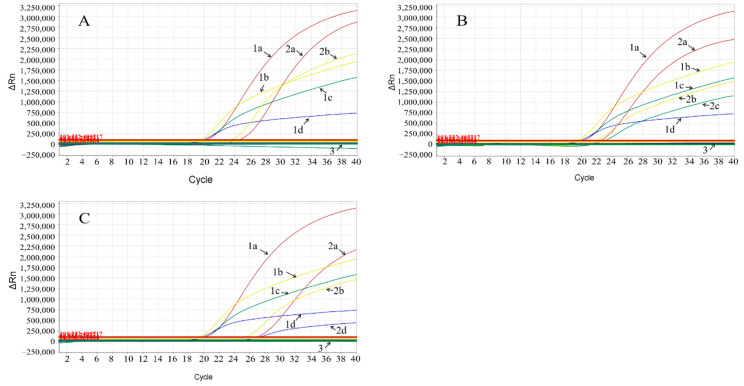
Evaluation of the clinical samples by the multiplex real-time qPCR: (**A**) ASFV-positive sample that lacked MGF505-2R and EP402R genes; (**B**) ASFV-positive sample that lacked EP402R gene; (**C**) ASFV-positive sample that lacked MGF505-2R gene. Curves 1a, 1b, 1c and 1d are the dynamic curves of the standard plasmids pASFV-B646L, pASFV-I177L, pASFV-MGF505-2R and pASFV-EP402R, respectively, which were used as positive controls. Curves 2a, 2b, 2c and 2d are the dynamic curves of the ASFV B646L, I177L, MGF505-2R and EP402R genes in the clinical samples, respectively. Curve 3: Negative control.

**Table 1 animals-12-01754-t001:** Primer and probe sequences used in this study.

Target Gene	Name	Sequences (5′→3′)	Amplicon (bp)
B646L	P72-F	GGCGTATAAAAAGTCCAGGAAATTC	79
	P72-R	TTCGGCGAGCGCTTTATC	
	P72-P	FAM-TCACCAAATCCTTTTGCGATGCAAGCT-BHQ1	
MGF505-2R	505-2R-F	AGTCATGCACGGCATATACAA	153
	505-2R-R	GGTTTAAACCGTGCCACATCC	
	505-2R-P	VIC-ACGCGGCCACCCAATTCAGAGAC-BHQ1	
EP402R	EP402R-F	TACTACATGCGTCCCTCAACAC	182
	EP402R-R	AATGGCGGGATATTGGGTAGT	
	EP402R-P	CY5-ACCGTGTCCTCCACCCAAACCAT-BHQ2	
I177L	I177L-F	GGCATAATTATCAAATGCGAAGGG	122
	I177L-R	TGGAAAGTTAATGATCAGGGCTT	
	I177L-P	Texas Red-AATCCTAGCTTGCCGGTAATGGCT-BHQ2	

**Table 2 animals-12-01754-t002:** The reaction system of the multiplex real-time qPCR and the OIE-recommended real-time qPCR.

Multiplex Real-Time qPCR			OIE-Recommended Real-Time qPCR		
Reagent	Volume (μL)	Final Concentration (nM)	Reagent	Volume (μL)	Final Concentration (nM)
Premix Ex Taq (2×)	10	1×	Premix Ex Taq (2×)	12.5	1×
P72-F (20 μM)	0.2	200	P72-P1 (50 μM)	1	2000
P72-R (20 μM)	0.2	200	P72-P2 (50 μM)	1	2000
P72-P (20 μM)	0.2	200	P72-P (5 μM)	1	200
I177L-F (20 μM)	0.1	100	/	/	/
I177L-R (20 μM)	0.1	100	/	/	/
I177L-P (20 μM)	0.1	100	/	/	/
EP402R-F (20 μM)	0.2	200	/	/	/
EP402R-R (20 μM)	0.2	200	/	/	/
EP402R-P (20 μM)	0.2	200	/	/	/
MGF505-2R-F (20 μM)	0.2	200	/	/	/
MGF505-2R-R (20 μM)	0.2	200	/	/	/
MGF505-2R-P (20 μM)	0.3	300	/	/	/
Template	2	/	Template	2.5	/
Distilled water	Up to 20	/	Distilled water	Up to 25	/

**Table 3 animals-12-01754-t003:** Repeatability and reproducibility analysis of the multiplex real-time qPCR.

Plasmid	Concentration (Copies/μL)	Ct Values of Intra-Assay for Repeatability			Ct Values of Inter-Assay for Reproducibility		
		$\bar{X}$	SD	CV (%)	$\bar{X}$	SD	CV (%)
pASFV-B646L	3.21 × 10^8^	12.60	0.10	0.80	12.69	0.13	1.01
	3.21 × 10^6^	18.45	0.16	0.88	18.45	0.11	0.62
	3.21 × 10^4^	24.53	0.18	0.71	24.75	0.30	1.20
pASFV-I177L	3.21 × 10^8^	12.41	0.07	0.59	12.45	0.10	0.84
	3.21 × 10^6^	18.42	0.17	0.92	18.37	0.13	0.69
	3.21 × 10^4^	24.44	0.23	0.92	24.37	0.20	0.83
pASFV-MGF505-2R	3.21 × 10^8^	12.57	0.09	0.62	12.59	0.17	1.36
	3.21 × 10^6^	18.70	0.17	0.89	18.75	0.21	1.10
	3.21 × 10^4^	24.81	0.11	0.43	24.79	0.16	0.63
pASFV-EP402R	3.21 × 10^8^	13.18	0.07	0.53	13.29	0.13	0.99
	3.21 × 10^6^	19.05	0.09	0.46	19.03	0.30	1.56
	3.21 × 10^4^	25.13	0.07	0.26	25.19	0.13	0.53

**Table 4 animals-12-01754-t004:** Evaluation of the clinical samples (positive/total) (percentage).

Source	Tissue	Environmental Swab	Whole Blood	Nasopharyngeal Swab	Total
Pig farm	/	2/107 (1.87%) ^c^	11/1134 (0.97%)	3/37 (8.11%)	15/1278 (1.25%) ^d^
Slaughterhouse	58/480 (12.08%) ^b^	214/752 (28.46%) ^a^	/	/	330/1908 (17.30%) ^b^
Farmer’s market	14/175 (8.00%) ^b^	53/451 (11.75%) ^b^	/	/	67/626 (10.70%) ^c^
Harmless disposal site	105/365 (28.77%) ^a^	16/62 (25.81%) ^a^	/	/	121/427 (28.34%) ^a^
Total	177/1020 (17.35%) ^B^	285/1372 (20.77%) ^A^	11/1134 (0.97%) ^D^	3/37 (8.11%) ^C^	534/4239 (12.60%)

Note: The data in the same column which are marked with different small letters indicate significant differences (*p* < 0.05). The data in the last line which are marked with different capital letters indicate significant differences (*p* < 0.05). The harmless disposal site is the place where dead pigs are concentrated from one or more different pig farms for harmless treatment.

## Data Availability

The data used in the analysis can be obtained from the authors on request.

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
