# Peer review of "The Development of a Multiplex Real-Time Quantitative PCR Assay for the Differential Detection of the Wild-Type Strain and the MGF505-2R, EP402R and I177L Gene-Deleted Strain of the African Swine Fever Virus"

_animals, 2022, doi:10.3390/ani12141754_

Round 1
Reviewer 1 Report
The current manuscript described the development of a multiplex real-time qPCR for the differential detection of wild-type and gene-deleted ASFV strains. Authors showed the specificity, sensitivity, and repeatability of the assay with a significant number of samples.
The manuscript is well written and and the experiments were well developed. I suggest some minor changes before publishing
Introduction
A novel vaccine has recently been released at the Vietnamese market, please, add the relevant information and update this topic in the introduction accordingly.
Material and methods
L99: add the name of the institution which provided the strain
To ease following the sample proceeding, I suggest to reorganized some of the subheads. 2.4. DNA Extraction should be described direct after 2.2. Clinical Samples and numbered accordingly. Additionally, the information about sample preparation and DNA extraction which is included in 2.11. Test of Clinical Samples by the Multiplex Real-time qPCR should be added in DNA Extraction accordingly.
L103: what matrices were obtained in harmless disposal sites?
L118: What matrices was obtained to test environmental samples?
L120: change to “positive DNA as template”
The subhead Test of Clinical Samples by the Multiplex Real-time qPCR should focus on the performed test.
In statistical analysis, I suggest to add the regression analysis and the Pearson’s correlation coefficient which was used to calculate the analytical sensibility.
Results
Please, add a table summarizing the PCR profile and primer volumes/concentrations described in 3.2. Optimal Reaction Conditions of the Multiplex Real-time qPCR
L243: remove the statement “indicating an excellent linear relationship between the initial templates and the Ct values” or described avoiding subjective statements when describing results.
L275: remove or rephrase the statement “demonstrating high sensitivity of the assay”
L294: Rephrase the statement “of different types of sources” for “out of the different sample types”
Discussion
L 332: add the relevant information about the released vaccine into the Vietnamese market
Author Response
4 July, 2022
Revision notes
Dear editor and reviewers,
We have made all efforts to meet the comments and suggestions provided by the editor and reviewers. We have revised our manuscript carefully according to the reviwers’s suggestions. Our responses are described in detail as follows.
Reviewer 1:
Introduction
- A novel vaccine has recently been released at the Vietnamese market, please, add the relevant information and update this topic in the introduction accordingly.
Response: The information on the Vietnamese vaccine has been added in the Introduction. Please see Lines 78-81 in the revised manuscript.
Material and methods
- L99: add the name of the institution which provided the strain
Response: The name of the institution has been added. Please see lines 108 in the revised manuscript.
- To ease following the sample proceeding, I suggest to reorganized some of the subheads. 4. DNA Extractionshould be described direct after 2.2. Clinical Samples and numbered accordingly. Additionally, the information about sample preparation and DNA extraction which is included in 2.11. Test of Clinical Samples by the Multiplex Real-time qPCR should be added in DNA Extraction accordingly.
Response: The subheads in the Materials and Methods have been reorganized. 2.4. DNA Extraction has been changed to 2.3. DNA Extraction and described directly after 2.2. Clinical Samples. In addition, the information about sample preparation and DNA extraction in 2.11. Test of Clinical Samples by the Multiplex Real-time qPCR has been added in 2.3 DNA Extraction. Please see lines 117-133 in the revised manuscript.
- L103: what matrices were obtained in harmless disposal sites?
Response: The harmless disposal site (plant) is the place where dead pigs are concentrated for harmless treatment. In these sites, the tissue samples and environmental samples were collected and tested for ASFV.
- L118: What matrices was obtained to test environmental samples?
Response: The swabs from environments of harmless disposal sites, slaughterhouses, pig farms, and markets were collected and tested for ASFV.
- L120: change to “positive DNA as template”
Response: Done. Please see line 146 in the revised manuscript.
- The subhead Test of Clinical Samples by the Multiplex Real-time qPCR should focus on the performed test.
Response: Done. This part focus on the performed test. Please see lines 208-241 in the revised manuscript: 2.11. Test of Clinical Samples by the Multiplex Real-time qPCR.
- In statistical analysis, I suggest to add the regression analysis and the Pearson’s correlation coefficient which was used to calculate the analytical sensibility.
Response: Done. Please see lines 243-246 in the revised manuscript.
Results
- Please, add a table summarizing the PCR profile and primer volumes/concentrations described in 2. Optimal Reaction Conditions of the Multiplex Real-time qPCR
Response: One table (Table 2) has been added to summarize the multiplex qPCR reaction system. Please see lines 239-241 Table 2 in the revised manuscript.
- L243: remove the statement “indicating an excellent linear relationship between the initial templates and the Ct values” or described avoiding subjective statements when describing results.
Response: The statement has been deleted. Please see lines 281-282 in the revised manuscript.
- L275: remove or rephrase the statement “demonstrating high sensitivity of the assay”
Response: The statement has been deleted. Please see line 312 in the revised manuscript.
- L294: Rephrase the statement “of different types of sources” for “out of the different sample types”
Response: Done. Please see line 331 in the revised manuscript.
Discussion
- L 332: add the relevant information about the released vaccine into the Vietnamese market
Response: This information was discussed in Discussion. Please see lines 416-423.
Reviewer 2:
- In my opinion, some language and style improvements are required to make the text more clear.
Response: The manuscript has been revised carefully according to the reviewers’s suggestions. Please see the revised manuscript.
- 48-49 ASF is not HIGHLYcontagious in fact
Response: delete “highly contagious”. Please see lines 48-49.
- 68-69 «Vaccination with effective vaccines is the most useful strategy for preventing and controlling ASF» - a disputable statement, it could be, but not definitely.
Response: This sentence has been revised to “Vaccination with effective vaccines is one of the most useful strategies for preventing and controlling ASF”. Please see lines 72-73.
- 72 – «man-made deletion» = recombinant strain?
Response: This sentence has been re-written. Please see lines 75-78 in the revised manuscript.
- 103 – what do you mean by “harmless disposal sites”?
Response: The harmless disposal site is the place where dead pigs are concentrated from one or more different pig farms for harmless treatment. A note has been added under Table 4. Please see lines 346-347.
Reviewer 3:
- What software was used to design the primers? Were they checked against primer-dimers? EP402R amplification product seems to be quite long for qPCR, this region has a high AT content leading to the troubles to design a useful PCR assay. Please, elaborate.
Response: The Primer Express Software v3.0 (ABI, USA) was used to design four pairs of specific primers and corresponding TaqMan probes. The primer-dimers and specificy of the primers, and the specificy of the probes were analyzed. This information has been added in the revised manuscript. Please see lines 135-136.
The primers were designed carefully to amplify 185 bp fragment of ASFV EP402R gene. The sequence of the fragment was showed as follows. This region is not high AT content.
Fragment sequence:
TACTACATGCGTCCCTCAACACAACCACTCAACCCATTTCCCTTACCTAAACCGTGTCCTCCACCCAAACCATGTCCGCCACCCAAACCATGTCCTCCACCTAAACCATGTCCTTCAGCTGAATCCTATTCTCCACCCAAACCACTACCTAGTATCCCGCTACTACCCAATATCCCGCCATT
Sequence aligement:
- The target regions were selected properly, however, how could you be sure that the assay is able to differentiate between deletion of whole gene (as it is likely in vaccine strains) and partial deletion (as it was found in naturally mutated strains?), moreover there is a possibility to miss partial mutation located within the same gene, but external to qPCR target site.
Response: In this study, four pairs of primers were designed to amplify partial region, but not whole region of B646L, MGF505-2R, EP402R, and I177L genes, respectively. If the targeted fragment of a gene could not be detected, the gene was considered to be deleted, but it could not verify whether the whole gene or part of the gene was deleted.
- Developed assay may be considered to be useful to detect mutated stains, however, it is impossible at this stage to confirm whether it was vaccine, or naturally occurring mutant. Please discuss the need to develop DIVA test assay to differentiate between vaccine strains and wild types.
Response: The multiplex qPCR developed in this study could differentiate the wild-type strain and the MGF505-2R, EP402R, and I177L gene-deleted strains. The assay could not confirm whether it was vaccine, or naturally occurring mutant. So far, no commercial vaccine has been officially approved for application in the field in China. Therefore, the gene-deleted strains detected in the clinical samples from pig farms without using vaccine were considered to be naturally gene-deleted strains. It needs further study to develop an assay to differentiate between vaccine strains and wild-type strains.
- One of the most intriguing part of the study was selection of various samples to detect ASFV. Except pig tissue and blood, also environmental swabs collected from slaughterhouses, pig farms, and markets were selected, which is not so obvious, at least according to European Union ASFV testing requirements. Is this sampling strategy validated and allowed in China and the samples originated from routine surveillance? Or You just selected them based on available literature data showing that they are also useful for ASFV detection? The most important thing here is high positivity rate of this environmental swabs, it must be better highlighted and discussed, because it indicates that this sampling strategy maybe should be employed in other countries to control and prevent disease transmission? Moreover, it also suggest that ASFV in China is still out of control, and pose a relevant risk to other countries.
Response: Since ASFV can survive for a long period in the environment, the viruses can easily spread from the polluted places through moving human and vehicles. Therefore, the environmental swabs collected from harmless disposal sites, slaughterhouses, pig farms, and markets were selected to test ASFV. This is a routine surveillance in order to assess the risk of environmental pollution on the spread of ASFV, which is a part of the plan of the Ministry of agriculture and rural areas of China to perform daily monitoring of ASFV.
Since the environmental samples were high positive for ASFV, it was highlighted to discussed in Discussion. Please see lines 402-408 in the revised manuscript.
The results showed that 12.60% (534/4239) samples were positive for ASFV, and of the 534 positive samples, 285 (53.37%) were environmental samples. ASF is still a serious disease.
- What is "harmless disposal site"?
Response: The harmless disposal site is the place where dead pigs are concentrated from one or more different pig farms for harmless treatment. A note has been added under Table 4. Please see lines 346-347.
- Line 373-374: Since you mentioned illegal vaccine strains which – according to media reports – are currently circulating in China, please relate to the ASFV genome low variability and low mutation rate, leading to low probability of such kind of variations.
Response: So far, no commercial vaccine has been officially approved for application in the field in China. Any ASF vaccine is illegal in China, and any attempt to use vaccines in China is prohibited until now. Therefore, the gene-deleted strains detected in the clinical samples from pig farms without using vaccine were considered to be naturally gene-deleted strains. However, it cannot be ruled out that some people take risks and use illegal vaccines.
Even if ASFV genome showed low variability and low mutation rate, the naturally gene-deleted strains were reported in China in 2020. Please see lines 510-512 reference 27.
Best regards,
Kaichuang Shi
Vice director, and Professor
Guangxi CADC, China

Reviewer 2 Report
Thank you for interesting reading and practically useful results of your work.
In my opinion, some language and style improvements are required to make the text more clear.
There are some comments on the text
48-49 ASF is not HIGHLY contagious in fact
68-69 «Vaccination with effective vaccines is the most useful strategy for preventing and controlling ASF» - a disputable statement, it could be, but not definitely
72 – «man-made deletion» = recombinant strain?
103 – what do you mean by “harmless disposal sites”?
Author Response

(The authors gave the same response as above.)

Reviewer 3 Report
The manuscript entitled “Development of a multiplex real-time quantitative PCR assay 2 for differential detection of the wild-type strain and the 3 MGF505-2R, EP402R, and I177L gene-deleted strain of African swine fever virus” describes the development of multiplex real-time PCR assay to differentiate wild-type from deleted strains of virus, which is currently circulating in China. Since the ASFV poses a serious concern for worldwide pig production, thus the subject is of particular importance. The study is interesting, well designed and properly performed. The conclusions are supported by the obtained data. Language is sufficient, however please check it again to improve some minor grammatical errors before any further proceedings. The study may be accepted for publication after development or at least discussion of the following issues:
1. What software was used to design the primers? Were they checked against primer-dimers? EP402R amplification product seems to be quite long for qPCR, this region has a high AT content leading to the troubles to design a useful PCR assay. Please, elaborate.
2. The target regions were selected properly, however, how could you be sure that the assay is able to differentiate between deletion of whole gene (as it is likely in vaccine strains) and partial deletion (as it was found in naturally mutated strains?), moreover there is a possibility to miss partial mutation located within the same gene, but external to qPCR target site.
3. Developed assay may be considered to be useful to detect mutated stains, however, it is impossible at this stage to confirm whether it was vaccine, or naturally occurring mutant. Please discuss the need to develop DIVA test assay to differentiate between vaccine strains and wild types.
4. One of the most intriguing part of the study was selection of various samples to detect ASFV. Except pig tissue and blood, also environmental swabs collected from slaughterhouses, pig farms, and markets were selected, which is not so obvious, at least according to European Union ASFV testing requirements. Is this sampling strategy validated and allowed in China and the samples originated from routine surveillance? Or You just selected them based on available literature data showing that they are also useful for ASFV detection? The most important thing here is high positivity rate of this environmental swabs, it must be better highlighted and discussed, because it indicates that this sampling strategy maybe should be employed in other countries to control and prevent disease transmission? Moreover, it also suggest that ASFV in China is still out of control, and pose a relevant risk to other countries.
5. What is "harmless disposal site"?
6. Line 373-374: Since you mentioned illegal vaccine strains which – according to media reports – are currently circulating in China, please relate to the ASFV genome low variability and low mutation rate, leading to low probability of such kind of variations occurring naturally.
Author Response

(The authors gave the same response as above.)
